# The Effect of Aerobic Exercise Training Frequency on Arterial Stiffness in a Hyperglycemic State in Middle-Aged and Elderly Females

**DOI:** 10.3390/nu13103498

**Published:** 2021-10-03

**Authors:** Ryota Kobayashi, Kenji Asaki, Takeo Hashiguchi, Hideyuki Negoro

**Affiliations:** 1Center for Fundamental Education, Teikyo University of Science, Tokyo 120-0045, Japan; 2Department of Tokyo Judo Therapy, Teikyo University of Science, Tokyo 120-0045, Japan; k-asaki@ntu.ac.jp; 3Department of School Education, Teikyo University of Science, Tokyo 120-0045, Japan; hasiguti@ntu.ac.jp; 4Harvard PKD Center, Boston, MA 02115, USA; oystercope@gmail.com; 5Faculty of Medicine, Nara Medical University, Nara 634-8521, Japan

**Keywords:** arterial stiffness, glucose ingestion, aerobic exercise, training sessions per week, middle-aged and elderly patients

## Abstract

The frequency of aerobic exercise training in reducing the increase in arterial stiffness during acute hyperglycemia, a risk factor for cardiovascular disease, is unknown. The aim of the study was to determine the aerobic exercise training frequency on arterial stiffness in a hyperglycemic state in middle-aged and elderly females. Twenty healthy elderly people were randomly assigned to a two-times-a-week (T2, n = 10) and four-times-a-week (T4, n = 10) exercise group. All participants exercised for 35 min per session, which consisted of jogging exercises with a heart rate intensity of 65%. Brachial-ankle (ba), and heart-brachial (hb) pulse wave velocity (PWV) were measured before, 4 and 8 weeks after intervention; before the oral ingestion of 75-g of glucose; and 30, 60, and 90 min after ingestion. The baPWV before and 4 weeks after the intervention increased in both groups (*p* < 0.05), but only increased 8 weeks after intervention in the T2 group. hbPWV was unchanged before, 4 and 8 weeks after intervention in both groups. These findings show that frequent aerobic exercise suppresses the increase in arterial stiffness following glucose intake. The results of this study can be used to support the implementation of exercise programs for middle-aged and elderly patients.

## 1. Introduction

Acute hyperglycemia is an independent risk factor for cardiovascular disease [1]. Postprandial blood glucose has a higher risk of cardiovascular disease than fasting blood glucose [2]. The reason that acute hyperglycemia increases the risk of cardiovascular disease is associated with a transient increase in arterial stiffness [3]. Gordin et al. [4] suggested that healthy middle-aged and elderly people have increased arterial stiffness and elevated postprandial blood glucose levels. Moreover, systemic arterial stiffness has been reported to increase after the ingestion of glucose (such as the amount in a soft drink) in middle-aged and elderly individuals [5]. Because arterial stiffness increases with age in older postmenopausal women, there is a significant association between arterial stiffness and postprandial blood glucose levels [6]. Tsuboi et al. [7] reported that blood glucose measured after a meal was associated with arterial stiffness in women aged 50–85 years, but not in women younger than 50 years. Therefore, as Japan is a super-aging society [8] and since arterial stiffness is higher in women than in men [9], it is important to suppress the progression of arteriosclerosis during acute hyperglycemia in elderly females.

It is well known that regular endurance training improves cardiovascular function, including arterial destiffening [10,11,12]. For instance, Tanahashi et al. [13] reported a decrease in arterial stiffness in postmenopausal women after 12 weeks compared to before moderate-intensity aerobic exercise training. Thus, aerobic exercise training may decrease arterial stiffness. To the best of our knowledge, the effect of aerobic exercise on arterial stiffness during acute hyperglycemia has not been fully investigated. In a previous study, it was found that elderly patients with aerobic exercise habits had no change in arterial stiffness during acute hyperglycemia [14]. It has been reported that increasing physical activity such as walking or jogging for four weeks can reduce the increase in arterial stiffness during acute hyperglycemia in middle-aged and elderly individuals [15]. Moreover, it has been suggested that moderate-intensity rather than low-intensity aerobic exercise and aerobic exercise before rather than after meals may reduce the increase in arterial stiffness during acute hyperglycemia [16,17]. Therefore, aerobic exercise to reduce the increase in arterial stiffness during acute hyperglycemia may require moderate-intensity, pre-prandial walking and jogging for at least four weeks.

As the effect of exercise intensity on the improvement of arterial stiffness by aerobic exercise in elderly females is limited and that the effect of exercise volume is reported to be greater [18]. A previous study reported that arterial stiffness was reduced only when exercise was performed twice a week rather than once a week [19]. Therefore, aerobic exercise should be performed at least twice per week to lower arterial stiffness. In addition, it has been reported that lower amounts of lifelong exercise training (four times a week for 30 min or more), consistent with current physical activity recommendations, elicit similar benefits against age-related arterial stiffness [20]. Thus, aerobic exercise four times a week throughout a person’s life might be associated with reduced arterial stiffness. Therefore, it is important to focus on a frequency of aerobic exercise of 2–4 sessions per week to reduce arterial stiffness. However, the frequency of aerobic exercise sessions per week that can reduce the increase in arterial stiffness during acute hyperglycemia is unknown. Therefore, it is necessary to examine whether two or four sessions of aerobic exercise per week, which are expected to reduce arterial stiffness, can suppress the increase in arterial stiffness during acute hyperglycemia.

This study hypothesized that aerobic exercise four times weekly reduces the increase in arterial stiffness during acute hyperglycemia in middle-aged and elderly females more than aerobic exercise twice weekly. To prove this hypothesis, the purpose of this study is to examine the effect of the frequency of aerobic exercise training on arterial stiffness in hyperglycemic conditions in middle-aged and elderly females.

## 2. Materials and Methods

### 2.1. Participants

The participants were all women and 20 healthy middle-aged and older adults. They were randomly divided into two groups by a random number generated by a computer with a one-in-two probability: a group that performed aerobic exercise training twice a week (n = 10; T2 group) and a group that performed aerobic exercise training four times a week (n = 10; T4 group). We collected participants by posting flyers for research cooperation to the residents near the Teikyo University of Science. In the end, we received 30 applicants, from which we selected 20 participants who met the following criteria. Participants who had abnormal blood test, urine test, diabetic (ADA/EASD diagnostic criteria), chest X-ray, or electrocardiogram (ECG) results during the year prior to the study and those who had problems with exercise (such as those with musculoskeletal injuries) were excluded from the study. Participation criteria for all participants were normotensive (Japanese standard: <140/90 mmHg), non-smokers with no apparent disease on ECG or other diagnostic tests, and had no exercise habits prior to the study according to a physical activity questionnaire (Table 1). A power analysis was performed using G* Power 3 to determine the appropriate sample size [21]. The magnitude of the effect of exercise on arterial stiffness was assumed to be 0.5. It was determined that each group should include nine participants to detect any differences with 80% power and a 5% two-sided alpha using analysis of variance. In this study, 10 participants were included in each group in consideration of dropout. This study was conducted in compliance with the Declaration of Helsinki on the basis of ethics, human rights, and the protection of the personal information of participants. Ethical approval for this study was obtained from the Ethics Committee of Teikyo University of Science (approval number 20A024). The study was also registered at the University Hospital Medical Information Network Center (UMIN Center) (UMIN Study No. UMIN000041634). All hard-copy (paper) study data were stored in a locked filing cabinet, and the electronic data were stored on a secured network drive with access granted only to those working within the research lab. This study was performed in accordance with the guidelines for human experimentation published by the institutional review committee.

### 2.2. Study Design

All participants were female due to twenty healthy, middle-aged, or elderly participants were randomly divided into aerobic exercise training groups that performed exercises twice per week (n = 10; T2 group; age, 66.3 ± 4.4 years; height, 157.6 ± 2.2 cm; weight, 56.4 ± 3.8 kg) or four times per week (n = 10; T4 group; age, 65.7 ± 3.3 years; height, 160.8 ± 2.3 cm; weight, 57.2 ± 0.9 kg). Exercise training consisted of 35 min of moderate-intensity (65% reserve heart rate (HR)) jogging under supervision. The weekly sessions continued for two months. All tests were conducted prior to the exercise intervention, after four weeks of exercise, and after eight weeks of exercises, as described below. The participants were instructed to maintain their normal diet and activities of daily living, except for exercise training, for the duration of the study. Intense physical activity (training and daily living activities), caffeine, and alcohol consumption were prohibited for 24 h prior to the experiment. The participants fasted beginning (10–12 h) at 9:00 pm on the day before the study began. Arterial stiffness of the systemic and proximal aorta, blood pressure (BP) of the upper arm and ankle, HR, and blood glucose (BG) levels were measured before (baseline) and 30, 60 and 90 min after the 75-g oral glucose tolerance test (75-g-OGTT). The participants rested in the supine position prior to each measurement. The maximal oxygen uptake was measured at the end of the measurements obtained 90 min after the 75-g-OGTT (Figure 1).

### 2.3. Body Composition

The participants’ heights were measured in 0.1 cm increments using a height meter. Bodyweight, body fat percentage, and BMI were measured to the nearest 0.1 kg using a precision instrument body composition analyzer (WB-150 PMA, Tanita, Tokyo, Japan).

### 2.4. Arterial Stiffness

The brachial ankle (ba) and heart brachial (hb) pulse wave velocities (PWVs) of all participants were measured using an automatic oscillometric device (PWV/ABI, Colin Medical Technology, Komaki, Japan), as previously described [22,23]. All measurements were obtained baseline and 30, 60, and 90 min after the 75-g-OGTT in the supine position in a quiet room. The daily coefficients of variation in our laboratory were 3 ± 1% and 3 ± 2% for the baPWV and hbPWV, respectively.

### 2.5. Upper Arm and Ankle Blood Pressure and Heart Rate

Brachial and ankle systolic BP (SBP), mean BP (MBP), diastolic BP (DBP), pulse pressure, and HR at rest were measured in the supine position using an automatic oscillometric PWV/ABI device (Omron-Colin, Tokyo, Japan) over the brachial and posterior tibial arteries [22]. All measurements were obtained baseline and 30, 60, and 90 min after the 75-g-OGTT in the supine position in a quiet room. The daily coefficients of variation in our laboratory were 2 ± 1%, 2 ± 2%, and 2 ± 1% for brachial BP, ankle BP and HR, respectively.

### 2.6. Blood Glucose

Venous blood was drawn from the participants’ left fingertips baseline and at 30, 60, and 90 min after glucose ingestion. BG was measured using the flavin–adenine dinucleotide glucose dehydrogenase method using a Glutest Neo Alpha glucometer (Sanwa Kagaku Kenkyusho, Tokyo, Japan) [15]. All measurements were obtained before and 30, 60, and 90 min after the 75-g-OGTT in the supine position in a quiet room. The daily coefficient of variation for the BG was 3 ± 1%.

### 2.7. Oral Glucose Tolerance Test

The 75-g-OGTT was conducted via the administration of Trelan^®^-G75 (Ajinomoto Pharmai Co. Ltd., Tokyo, Japan) after an overnight (10–12 h) fast. Following the diabetic guidelines, the participants consumed the glucose drink (225 mL) in a sitting position over a five-minute period [24].

### 2.8. Maximal Oxygen Uptake

Maximal oxygen uptake, an index of aerobic exercise capacity, was measured using a bicycle ergometer (Medergo EM-400, OG Wellness Co., Okayama, Japan) and oxygen uptake was monitored for each breath (VO2Master MW-1100, VO2 Master Health, location). After a one-minute warm-up at 15 W, the work rate was increased by 20 W for males and 15 W for females every minute. The HR was recorded every five seconds (OH1+, Polar Electro OY, Kempele, Finland). The subjective exercise intensity (Borg scale) and SBP were measured at the end of each one-minute stage.

### 2.9. Aerobic Exercise Training

The exercise classes were conducted on a walking trail near Adachi Ward from December 2020 to March 2021. Both groups underwent two or four 35-min sessions of moderate-intensity training per week for eight weeks. Participants in both groups participated in an eight-week exercise class during which they performed preparatory exercises (stretching) for five minutes followed by jogging for 30 min. These exercise classes were conducted under the guidance of a researcher who was trained in the exercise instruction methods. During the exercise sessions, an exercise HR monitor (ForeAthlete 45S, GARMIN Ltd., Schaffhausen, Switzerland) was worn by each participant on her left hand and the participants were instructed to walk at a speed at which their heart rate was 65% of their estimated maximum HR based on age. The estimated maximum heart rate based on age was calculated by subtracting age from the constant 220 and was set to 65% exercise intensity by the Carbonen method. In addition, the subjective exercise intensity was assessed every three minutes during the exercise session using the Borg scale. To monitor the participants’ daily activities during the study period and to help the participants avoid changes to their daily activities, an activity diary was completed.

### 2.10. Statistical Analysis

The data are presented as mean ± standard deviation (SD). Data normality and homogeneity of variance were investigated using the Shapiro–Wilk test and the Levene test, respectively. The changes of each measurement before and after the intervention are presented as mean values with 95% confidence intervals for the groups. Parametric analysis was a two-way analysis of variance using repeated measures (time * group) for the three measurements performed. Whenever the sphericity assumption was violated (Mauchly’s test), analyses were adjusted using Greenhouse–Geisser correction. The Bonferroni method was used for changes in each intervention using post hoc testing. SPSS (version 25, manufacturer name, location) was used for the statistical analyses. Statistical significance was set at α = 0.05, and all α were two-sided. Effect size (ES) was calculated to investigate the magnitude of differences based on Cohen’s d.

## 3. Results

### 3.1. Physical Characteristics

The physical characteristics of the T2 group were age 66.3 ± 4.4 years; height 157.6 ± 2.2 cm; weight 56.4 ± 3.8 kg, and the physical characteristics of the T4 group were age 65.7 ± 3.3 years; height 160.8 ± 2.3 cm; weight 57.2 ± 0.9 kg. The average patient age, height, weight, body fat percentage, and body mass index were not significantly different between the groups (Table 1). The mean weight, body fat percentage, and body mass index did not change within either group after four weeks of exercise or after eight weeks of exercise compared to baseline.

### 3.2. Arterial Stiffness

Pre-of exercises, the baPWV increased at 30 (*p* < 0.05), 60 (*p* < 0.01) and 90 min (*p* < 0.01) after the 75-g-OGTT in the T2 group. There was no significant difference in the comparison at other times. The baPWV increased at 30 (*p* < 0.05), 60 (*p* < 0.01) and 90 min (*p* < 0.01) after the 75-g-OGTT in the T4 group. There was no significant difference in the comparison at other times. There were no significant differences between the two groups (Figure 2A). 

After four weeks of exercises, the baPWV increased at 30 (*p* < 0.05), 60 (*p* < 0.05) and 90 min (*p* < 0.01) after the 75-g-OGTT in the T2 group. There was no significant difference in the comparison at other times. The baPWV was unchanged at 30 and 60 min after the 75-g-OGTT and increased at 90 min (*p* < 0.05) in the T4 group. There was no significant difference in the comparison at other times. There were no significant differences between the two groups (Figure 2A). 

After eight weeks of exercise, the baPWV increased at 30 (*p* < 0.05) and 60 min (*p* < 0.05) after 75-g-OGTT and remained unchanged at 90 min in the T2 group. There was no significant difference in the comparison at other times. The baPWV was unchanged at 30, 60, and 90 min after the 75-g-OGTT in the T4 group. There was no significant difference in the comparison at other times. There were no significant differences between the two groups. (Figure 2A).

There were no significant intra- or intergroup differences in the hbPWV at pre or after four or eight weeks of exercise during the 75-g-OGTT (Figure 2B).

### 3.3. HR

There were no significant differences in HR during the 75-g-OGTT at pre, after four weeks of exercise, or after eight weeks of exercise in either group (Table 2). The HR was not significantly different between the groups at any time point.

### 3.4. Brachial Blood Pressure

Pre-of exercises, the brachial SBP was unchanged at 30 and 60 min after the 75-g-OGTT and increased at 90 min (*p* < 0.05) in the T2 group. There was no significant difference in the comparison at other times. The brachial SBP was unchanged at 30 and 60 min after the 75-g-OGTT and increased at 90 min (*p* < 0.05) in the T4 group. The brachial MBP and DBP were unchanged at 30, 60, and 90 min after the 75-g-OGTT in the T2 group. There was no significant difference in the comparison at other times. The brachial MBP and DBP were unchanged at 30, 60, and 90 min after the 75-g-OGTT in the T4 group. There was no significant difference in the comparison at other times. There were no significant differences between the two groups (Table 2).

After four and eight weeks of exercises, the brachial SBP, MBP and DBP were unchanged at 30, 60, and 90 min after the 75-g-OGTT in the T2 group. There was no significant difference in the comparison at other times. The brachial SBP, MBP, and DBP were unchanged at 30, 60, and 90 min after the 75-g-OGTT in the T4 group. There was no significant difference in the comparison at other times. There were no significant differences between the two groups (Table 2).

### 3.5. Ankle Blood Pressure

Pre-of exercises, the ankle SBP increased at 30 (*p* < 0.05), 60 (*p* < 0.05) and 90 min (*p* < 0.01) after the 75-g-OGTT in the T2 group. There was no significant difference in the comparison at other times. The ankle SBP increased at 30 (*p* < 0.05), 60 (*p* < 0.05), and 90 min (*p* < 0.01) after the 75-g-OGTT in the T4 group. There was no significant difference in the comparison at other times. There were no significant differences between the two groups (Table 3). 

After four and eight weeks of exercise, the ankle SBP was unchanged at 30, 60, and 90 min after the 75-g-OGTT in the T2 group. There was no significant difference in the comparison at other times. The ankle SBP was unchanged at 30, 60 and 90 min after the 75-g-OGTT in the T4 group. There was no significant difference in the comparison at other times. There were no significant differences between the two groups (Table 3).

Pre, after four and eight weeks of exercise, the ankle MBP and DBP were unchanged at 30, 60, and 90 min after the 75-g-OGTT in the T2 group. There was no significant difference in the comparison at other times. The ankle MBP and DBP were unchanged at 30, 60, and 90 min after the 75-g-OGTT in the T4 group. There was no significant difference in the comparison at other times. There were no significant differences between the two groups (Table 3). 

### 3.6. BG

Pre-of exercises, the BG was increased at 30 (*p* < 0.01) and 60 min (*p* < 0.01) after the 75-g-OGTT in the T2 group. There was no significant difference in the comparison at other times. The BG was increased at 30 (*p* < 0.01) and 60 min (*p* < 0.05) after the 75-g-OGTT in the T4 group. There was no significant difference in the comparison at other times. There were no significant differences between the two groups (Figure 3).

After four and eight weeks of exercises, the BG was increased at 30 (*p* < 0.01) and 60 min (*p* < 0.01) after the 75-g-OGTT in the T2 group. There was no significant difference in the comparison at other times. The BG was increased at 30 min (*p* < 0.05) after the 75-g-OGTT in the T4 group. There was no significant difference in the comparison at other times. The BG was significantly lower in the T4 group than in the T2 group 30 min after the 75-g-OGTT (Figure 3).

### 3.7. Maximal Oxygen Uptake

In group T2, the maximal oxygen uptake increased after four weeks (*p* < 0.05) and eight weeks (*p* < 0.01) of exercise compared to pre (Table 4). In group T4, the maximal oxygen uptake increased after four weeks (*p* < 0.01) and eight weeks (*p* < 0.01) of exercise compared to pre. There were no significant differences in maximal oxygen uptake between the groups at any time point (Table 4).

## 4. Discussion

The results of this study indicate that aerobic exercise four times per week in the smallest number of participants can suppress the increase in arterial stiffness induced during hyperglycemia in older adults more than aerobic exercise two times per week. Therefore, the increase in arterial stiffness during acute hyperglycemia may be reduced in elderly patients who perform moderate-intensity aerobic exercises before meals four days per week for eight weeks. This reduction may be related to the frequency of exercise.

Systemic arterial stiffness is known to increase during acute hyperglycemia [14]. A previous study reported an increase in the baPWV, an index of systemic arterial stiffness, after the ingestion of a 25-g glucose solution in healthy, elderly individuals [5]. Similar results were obtained in the present study. In elderly individuals, the baPWV increased significantly 30 min after the 75-g-OGTT. Therefore, in elderly people, systemic arterial stiffness may increase during acute hyperglycemia. In contrast, the hbPWV, an indicator of proximal aortic stiffness, may not change during acute hyperglycemia in healthy individuals. Previous studies have reported that the hbPWV does not change after the ingestion of glucose in healthy, elderly participants [15], which is consistent with the results of this study where there was no change in the hbPWV after the 75-g-OGTT in healthy, middle-aged, and elderly participants. However, Baynard et al. [25] reported that aortic stiffness increased after a high-carbohydrate meal in obese individuals. Therefore, changes in aortic stiffness during acute hyperglycemia may be higher in obese patients with high insulin resistance. The results of this study also suggest that the increase in systemic arterial stiffness during acute hyperglycemia may be influenced by peripheral arterial stiffness rather than aortic stiffness. A previous study found that the femoral-ankle PWV, a marker of lower extremity arterial stiffness, increased in healthy participants during acute hyperglycemia, while the carotid-femoral PWV (cfPWV) did not change [16]. Furthermore, Gordin et al. [4] found that the baPWV, but not cfPWV, increased during acute hyperglycemia. In the present study, the baPWV increased after the 75-g-OGTT, but hbPWV did not change. Therefore, even in healthy individuals, the peripheral arteries may be preferentially hardened during hyperglycemia. To clarify these results, it is necessary to compare changes by site (aorta, proximal aorta, brachial artery, and lower limb artery) during acute hyperglycemia in elderly participants. 

Arterial stiffness increases in an age-dependent manner, but there are marked sex differences, with progression occurring after menopause in women. The postmenopausal progression of baPWV in women acts on cardiovascular disease risk [26]. Because arterial stiffness increases with age in older postmenopausal women, there is a significant association between arterial stiffness and postprandial blood glucose levels [6]. In fact, Tsuboi et al. [7] reported that blood glucose measured after a meal was associated with arterial stiffness in women aged 50–85 years, but not in women younger than 50 years. The age of the subjects in this study was 66.3 ± 4.4 years for the T2 group and 65.7 ± 3.3 for the T4 group. This means that postmenopausal women may have increased arterial stiffness during acute hyperglycemia. This means that there is a social need to control the increase in arterial stiffness in postmenopausal women. The results obtained in this study are considered to be of great clinical significance in that aerobic exercise training four days a week before meals may contribute to the prevention of atherosclerosis in postmenopausal women.

It is currently unclear whether aerobic exercise can suppress the increase in arterial stiffness during acute hyperglycemia. Previous studies suggest that the baPWV increases after glucose ingestion in older adults with low levels of physical activity, but not in older adults with high levels of physical activity [14]. Furthermore, it has been reported that increasing the amount of daily physical activity (including aerobic exercise) in elderly patients suppresses the increase in the baPWV during acute hyperglycemia [15]. In addition, the daily moderate-intensity physical activity time for the control group (middle and elderly people) with increased arterial stiffness during hyperglycemia was 10 min, which translates to 70 min per week [15]. In the present study, the total amount of aerobic exercise in the T2 group was similar at 60 min per week and the baPWV did not change before and after the 75-g-OGTT in the T4 group after eight weeks of exercise. However, in the T2 group, the increase in the baPWV during acute hyperglycemia could not be suppressed. Therefore, increasing the number of aerobic exercises per week (total time) may be necessary to control the increase in arterial stiffness during hyperglycemia.

The association of exercise and arterial stiffness during acute hyperglycemia has three possible mechanisms. First, SBP may be involved, as arterial stiffness has been associated with changes in SBP [27]. According to a previous study, lower extremity arterial stiffness and SBP increase in healthy participants after a meal [17]. In the present study, an increase in the baPWV and ankle SBP was observed after the 75-g-OGTT. However, Augustine et al. [28] reported that lower extremity arterial stiffness and SBP did not change after a meal in participants who exercised regularly. Similarly, in the present study, the baPWV and lower extremity SBP did not change after the 75-g-OGTT after eight weeks of exercise training. Therefore, the increase in arterial stiffness during hyperglycemia may be related to an increase in SBP. However, in the present study, the increase in ankle SBP during acute hyperglycemia was suppressed after the exercise program in both groups, though the increase in the baPWV was not suppressed in the T2 group, suggesting that factors other than blood pressure may be involved in the mechanism of suppressing the increase in arterial stiffness during acute hyperglycemia, including BG. Changes in arterial stiffness during hyperglycemia have also been associated with BG levels [29]. Increases in arterial stiffness and BG were reported after the ingestion of glucose in a previous study [7]. Arterial stiffness has been reported to be correlated with BG during hyperglycemia [29]. In the current study, the baPWV and BG increased later in the T2 group after the 75-g-OGTT after the exercise program, but did not change in the T4 group. Furthermore, BG levels 30 min after the 75-g-OGTT after eight weeks of intervention were lower in the T4 group than in the T2 group (147.1 ± 10.0 vs. 115.8 ± 6). Therefore, the suppression of elevated BG may play a role in the suppression of the increase in arterial stiffness during hyperglycemia in the T4 group. Previous studies have reported that hyperglycemia after glucose ingestion decreases blood flow in the lower limbs [30]. Furthermore, glucose uptake has been found to increase only in the legs of individuals that have undergone aerobic exercise training [31]. Taken together, these results suggest that there is a relationship between blood flow and glucose uptake in the lower limbs. In the present study, four sessions of aerobic exercise per week prevented the increase in arterial stiffness during acute hyperglycemia by slowing down the increase in the BG level after the 75-g-OGTT, increasing blood flow in the lower limbs and improving the bioavailability of nitric oxide, which may have prevented the increase in arterial stiffness during acute hyperglycemia. However, blood flow and nitric oxide levels were not measured in this study. 

Acute hyperglycemia is an independent risk factor for cardiovascular disease [32]. In addition, glucose-loaded 2-h blood glucose levels (postprandial hyperglycemia) are associated with a higher risk of cardiovascular disease than fasting blood glucose levels (chronic hyperglycemia) [2]. Thus, even among those who have not been diagnosed with diabetes, only high postprandial blood glucose levels are seen in some individuals, which may indicate a higher risk of cardiovascular disease. Hyperglycemia “per se” is a condition that exacerbates cardiovascular disease in acute coronary syndromes in individuals without pre-existing diabetes [33,34]. Hyperglycemia is associated with more severe microvascular occlusion in both diabetics and non-diabetics, but is more pronounced in diabetics [35]. The definition of stress hyperglycemia varies slightly from article to article and the presence of this acute finding is not necessarily the same as the diagnosis of diabetes. Acute hyperglycemia is a physiological response to hormones such as epinephrine and cortisol, which increase oxidative stress and affect vascular endothelial cell and tissue damage [36]. Previous studies suggest that acute hyperglycemia may not be a direct mediator of ischemic or reperfusion injury, but a surrogate for the ischemic zone [36]. Therefore, repeated bouts of acute hyperglycemia increase the risk of coronary artery disease and, as diabetes progresses, may lead to the development of microvascular disease due to blockage of small arteries and macrovascular disease due to blockage of large arteries.

This study was not without limitations. A limitation of the present study is the relatively small number of participants. We examined middle-aged and elderly postmenopausal women; thus, our findings would not be generalizable to different populations. In addition, insulin levels, endothelial function, and leg arterial stiffness were not measured, though these parameters may have an important effect on arterial stiffness. There were no control groups in this study. However, in a previous study, participants in the control group (middle-aged and elderly adults) with increased arterial stiffness during hyperglycemia performed 10 min of moderate-intensity physical activity per day (70 min per week) [15]. In the present study, the participants in the T2 group performed 60 min of exercise per week. Therefore, the total amount of aerobic exercise performed when the sessions were conducted twice per week in this study may not have been sufficient.

## 5. Conclusions

Medium-intensity aerobic exercise training four times per week for eight weeks suppressed the increase in the baPWV associated with elevated BG after glucose intake. The duration, intensity, and timing of aerobic exercise training are important, and the frequency of exercise may also need to be considered to prevent the increase in systemic arterial stiffness during acute hyperglycemia.

## Figures and Tables

**Figure 1 nutrients-13-03498-f001:**
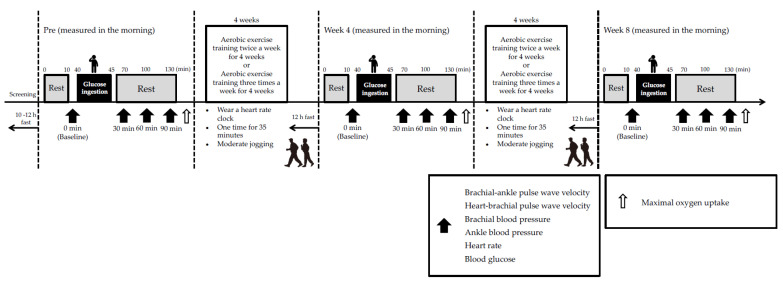
Study design. Arterial stiffness, blood pressure, heart rate, and blood glucose levels were measured baseline, 30, 60 and 90 min after the 75-g oral glucose tolerance test after 10 min of supine rest before, 4 and 8 weeks after the aerobic exercise training intervention. Thereafter, maximal oxygen uptake was measured using a bicycle ergometer.

**Figure 2 nutrients-13-03498-f002:**
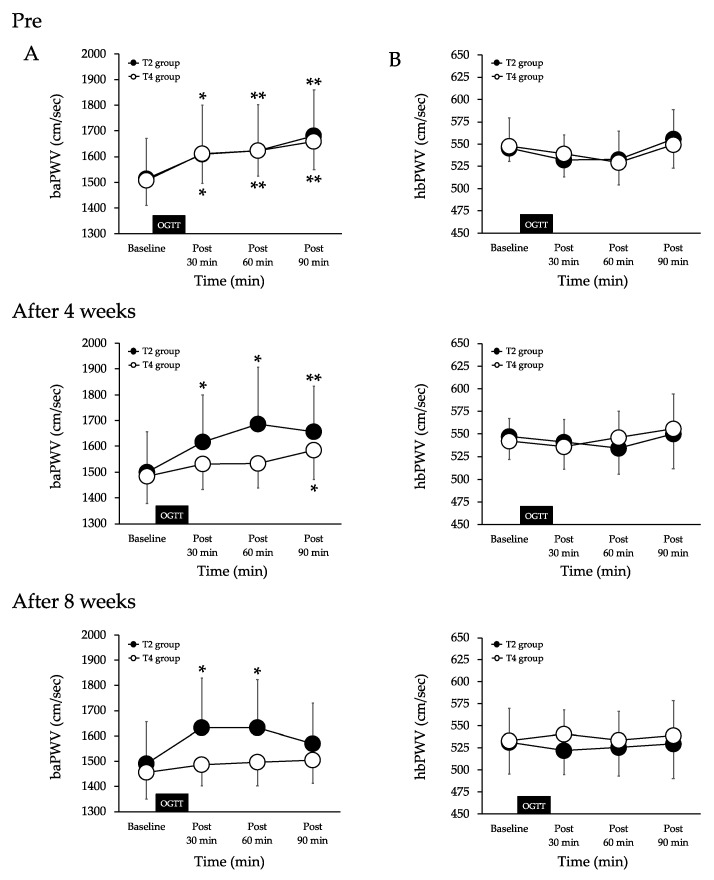
Changes in systemic (**A**) and proximal aortic stiffness (**B**) baseline and after the 75*-*g*-*OGTT in both groups. Values are mean ± SD. baPWV, brachial-ankle pulse wave velocity. hbPWV, heart-brachial pulse wave velocityT2 group, aerobic exercise training twice a week. T4 group, aerobic exercise training four times a week. OGTT, oral glucose tolerance test. ** *p* < 0.01 and * *p* < 0.05 vs. baseline.

**Figure 3 nutrients-13-03498-f003:**
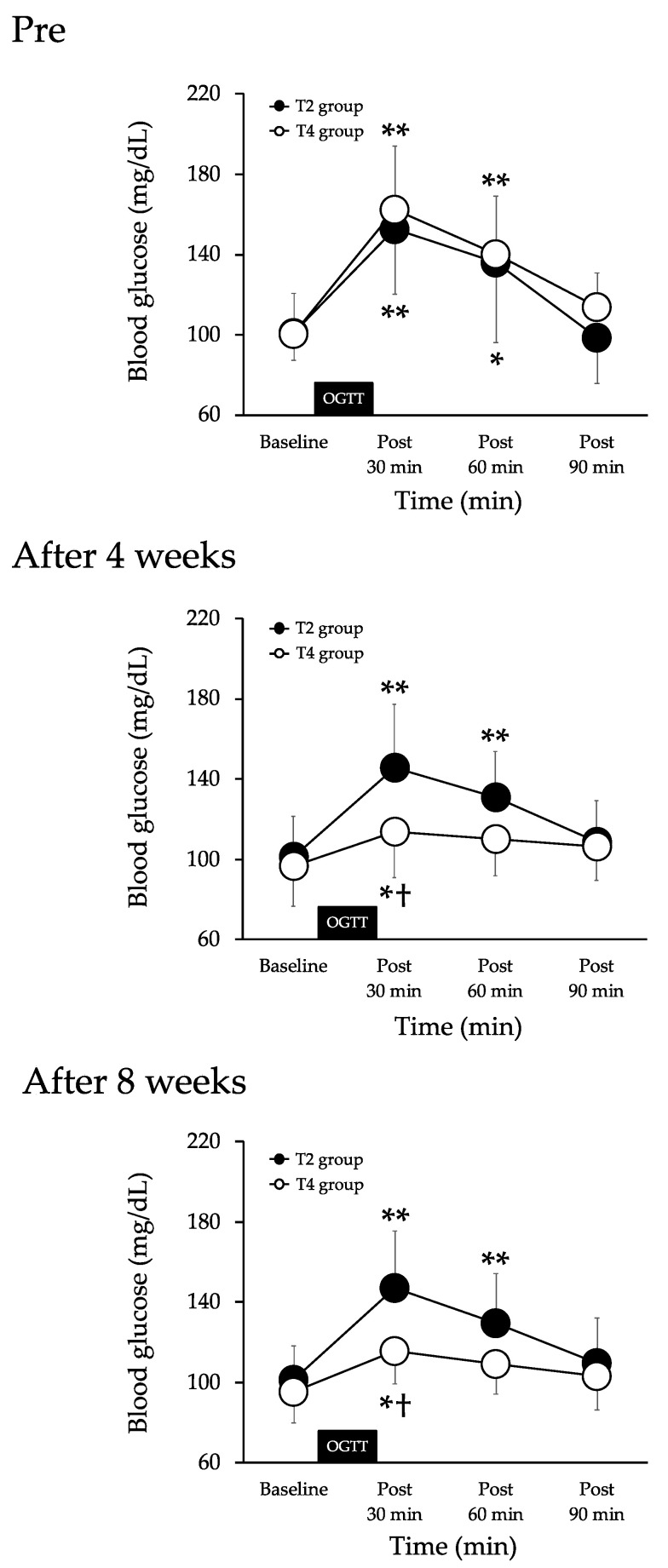
Changes in blood glucose before (baseline) and after the 75*-*g*-*OGTT in both groups. Values are mean ± SD. T2 group, aerobic exercise training twice a week. T4 group, aerobic exercise training 4 times a week. OGTT, 75-g oral glucose tolerance test. ** *p* < 0.01 and * *p* < 0.05 vs. baseline. ^†^ *p* < 0.05 vs. T2 group.

**Table 1 nutrients-13-03498-t001:** Summarizes the characteristics of both groups.

Variable	T2 Group			T4 Group			*p*-Value (Group)
	Pre	After 4 Week	After 8 Week	Pre	After 4 Week	After 8 Week	*p*-Value
Age, years	66.3 ± 4.4	N/A	N/A	65.7 ± 3.3	N/A	N/A	N/A
Height, cm	157.6 ± 2.2	N/A	N/A	160.8 ± 2.3	N/A	N/A	N/A
Weight, kg	56.4 ± 3.8	56.8 ± 4.0	56.0 ± 3.5	57.2 ± 0.9	57.3 ± 0.6	57.3 ± 0.4	0.308
BMI, kg/m^2^	22.7 ± 1.5	22.9 ± 1.6	22.6 ± 1.4	23.1 ± 0.4	23.1 ± 0.4	23.1 ± 0.4	0.318

Values are mean ± SD. BMI, body mass index. T2 group, aerobic exercise training twice a week. T4 group, aerobic exercise training 4 times a week.

**Table 2 nutrients-13-03498-t002:** Changes in brachial SBP, MBP and DBP before and after the 75-g OGTT of both groups.

Variable	Group	Intervention	Baseline	Post 30 min	Post 60 min	Post 90 min	*p*-Value (Group)
Brachial SBP, mmHg	T2 group	Pre	124.8 ± 5.9	128.1 ± 6.9	131.0 ± 8.1	132.0 ± 7.2 *	0.93
	T4 group		124.6 ± 7.0	126.0 ± 6.2	129.1 ± 5.8	133.0 ± 6.4	
	T2 group	After 4 weeks	122.9 ± 6.6	124.6 ± 7.3	128.1 ± 8.1	127.9 ± 7.8	0.93
	T4 group		123.1 ± 7.6	124.0 ± 6.1	125.5 ± 6.3	127.4 ± 4.8	
	T2 group	After 8 weeks	122.3 ± 5.1	124.8 ± 5.3	127.8 ± 6.7	126.5 ± 6.7	0.82
	T4 group		122.4 ± 4.5	122.7 ± 5.0	123.0 ± 6.1	125.9 ± 6.7	
Brachial MBP, mmHg	T2 group	Pre	90.3 ± 4.2	91.4 ± 4.2	92.9 ± 4.7	93.2 ± 4.4	0.85
	T4 group		89.7 ± 2.5	90.5 ± 2.5	91.3 ± 3.0	92.6 ± 2.5	
	T2 group	After 4 weeks	88.9 ± 4.7	89.5 ± 4.7	90.4 ± 4.8	90.8 ± 5.1	0.88
	T4 group		88.2 ± 3.4	88.5 ± 3.6	88.9 ± 3.8	90.6 ± 2.8	
	T2 group	After 8 weeks	88.1 ± 3.8	88.5 ± 3.4	90.1 ± 3.8	91.3 ± 4.1	0.81
	T4 group		87.4 ± 2.0	88.1 ± 2.1	88.5 ± 3.3	89.6 ± 2.8	
Brachial DBP, mmHg	T2 group	Pre	73.0 ± 3.5	73.1 ± 2.9	73.8 ± 3.3	73.8 ± 3.4	0.80
	T4 group		72.2 ± 2.3	72.7 ± 2.0	72.4 ± 2.9	72.4 ± 2.6	
	T2 group	After 4 weeks	71.9 ± 3.9	71.9 ± 3.9	71.5 ± 3.2	72.2 ± 3.9	0.86
	T4 group		70.8 ± 2.7	70.8 ± 3.2	70.6 ± 4.0	72.2 ± 3.0	
	T2 group	After 8 weeks	71.0 ± 3.2	70.4 ± 2.7	71.2 ± 2.5	73.7 ± 3.0	0.84
	T4 group		69.9 ± 2.3	70.7 ± 2.3	71.2 ± 3.2	71.5 ± 2.4	
HR, beats/min	T2 group	Pre	66.1 ± 3.3	64.0 ± 2.0	63.1 ± 2.6	62.3 ± 2.0	0.50
	T4 group		63.6 ± 3.4	62.9 ± 2.9	59.9 ± 2.5	59.4 ± 2.8	
	T2 group	After 4 weeks	65.9 ± 3.6	64.9 ± 2.5	63.3 ± 1.8	60.8 ± 2.6	0.18
	T4 group		62.7 ± 4.8	59.4 ± 2.7	57.1 ± 2.2	55.4 ± 1.8	
	T2 group	After 8 weeks	62.3 ± 3.4	61.5 ± 3.1	62.4 ± 2.2	61.1 ± 2.7	0.35
	T4 group		59.4 ± 2.4	60.7 ± 2.1	56.9 ± 1.7	57.1 ± 1.8	

Values are mean ± SD; T2 group, aerobic exercise training twice a week; T4 group, aerobic exercise training four times a week; 75-g OGTT, 75-g oral glucose tolerance test; SBP, systolic blood pressure; MBP, mean blood pressure; DBP, diastolic blood pressure; HR, heart rate. * *p* < 0.05 vs. baseline.

**Table 3 nutrients-13-03498-t003:** Changes in ankle SBP, MBP and DBP before and after the 75-g OGTT of both groups.

Variable	Group	Intervention	Baseline	Post 30 min	Post 60 min	Post 90 min	*p*-Value (Group)
Ankle SBP, mmHg	T2 group	Pre	149.2 ± 7.5	155.6 ± 7.8 *	162.1 ± 8.7 *	163.2 ± 8.9 **	0.98
	T4 group		150.2 ± 12.2	156.6 ± 11.6 *	161.6 ± 13.9 *	163.0 ± 10.7 **	
	T2 group	After 4 weeks	149.9 ± 7.5	152.1 ± 6.5	151.8 ± 6.9	153.4 ± 5.3	0.92
	T4 group		48.8 ± 13.4	152.7 ± 10.8	152.9 ± 11.1	154.4 ± 11.5	
	T2 group	After 8 weeks	149.0 ± 8.6	149.2 ± 7.0	151.7 ± 8.8	153.9 ± 7.5	0.97
	T4 group		147.2 ± 9.2	150.2 ± 9.7	153.1 ± 10.3	152.1 ± 10.3	
Ankle MBP, mmHg	T2 group	Pre	98.6 ± 4.4	101.0 ± 4.3	103.1 ± 5.0	104.7 ± 5.1	0.95
	T4 group		98.9 ± 4.1	101.5 ± 4.0	102.7 ± 4.3	105.6 ± 3.1	
	T2 group	After 4 weeks	96.6 ± 3.8	97.5 ± 3.5	97.6 ± 4.0	98.3 ± 3.9	0.98
	T4 group		97.1 ± 5.1	98.4 ± 3.8	98.4 ± 3.8	100.1 ± 3.8	
	T2 group	After 8 weeks	97.2 ± 5.4	97.0 ± 4.2	98.7 ± 4.9	100.6 ± 4.7	0.95
	T4 group		95.8 ± 2.8	96.7 ± 2.9	98.0 ± 3.3	99.9 ± 3.5	
Ankle DBP, mmHg	T2 group	Pre	73.4 ± 3.1	73.8 ± 2.7	73.6 ± 3.4	75.4 ± 3.6	0.93
	T4 group		73.3 ± 1.9	73.9 ± 1.2	73.3 ± 1.9	76.9 ± 2.8	
	T2 group	After 4 weeks	70.0 ± 2.2	70.2 ± 2.2	70.5 ± 2.7	70.7 ± 3.4	0.91
	T4 group		71.3 ± 2.2	71.2 ± 2.3	71.2 ± 2.4	73.0 ± 2.1	
	T2 group	After 8 weeks	71.3 ± 3.9	70.9 ± 3.1	72.3 ± 3.1	73.9 ± 3.5	0.84
	T4 group		70.1 ± 2.0	69.6 ± 1.9	70.4 ± 2.3	73.7 ± 1.7	

Values are mean ± SD; T2 group, aerobic exercise training twice a week; T4 group, aerobic exercise training four times a week; 75-g OGTT, 75-g oral glucose tolerance test; SBP, systolic blood pressure; MBP, mean blood pressure; DBP, diastolic blood pressure; HR, heart rate. ** *p* < 0.01 and * *p* < 0.05 vs. baseline.

**Table 4 nutrients-13-03498-t004:** Changes in VO_2_max in both groups.

Variable	T2 Group			T4 Group			*p*-Value (Group)
	Pre	After 4 Week	After 8 Week	Pre	After 4 Week	After 8 Week	*p*-Value
VO_2_max, ml/kg/min	21.4 ± 1.4	23.4 ± 1.3 *	25.2 ± 1.5 **	22.6 ± 2.8	27.0 ± 2.5 **	30.1 ± 1.4 **	0.204

Values are mean ± SD. VO_2_max, maximal oxygen uptake. T2 group, aerobic exercise training twice a week. T4 group, aerobic exercise training 4 times a week. ** *p* < 0.01 and * *p* < 0.05 vs. pre.

## Data Availability

The data presented in this study are available on request from the corresponding author.

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
