# Peer review of "The Effect of Aerobic Exercise Training Frequency on Arterial Stiffness in a Hyperglycemic State in Middle-Aged and Elderly Females"

_nutrients, 2021, doi:10.3390/nu13103498_

Round 1

Reviewer 1 Report

This study conducted on a small group of subjects is interesting and quite original. However, this reviewer raises some issues that have to be addressed by authors.

Major comments

1- Authors must add diabetes in the exclusion criteria. They probably did not do so by considering it implicit.

2- Moreover, if the patients were not diabetic, were all ADA/EASD diagnostic criteria used to rule out diabetes?

3- The sample size of this study is very small indeed. This issue, in addition to being stressed perhaps with more emphasis in the limitations section of the study, should also be adequately commented at the beginning of the discussion.

4- Correctly, the authors stated ‘Systemic arterial stiffness is known to increase during acute hyperglycemia’ as well as ‘Arterial stiffness is an important determinant of cardiovascular risk’. Actually, hyperglycemia "per se" is a condition that worsens CV outcome during ACS in individuals with no known diabetes (Journal of Clinical Endocrinology and Metabolism Volume 97, Issue 3, March 2012, Pages 933-942. doi: 10.1210/jc.2011-2037 - Journal of Diabetes Research, 2018, art. no. 3106056. doi: 10.1155/2018/3106056). Both this clinically very important issue and above referenced papers should be commented upon in the discussion.

Minor comments

1- Figure 1 is too small and hard to read.

2- The manuscript should be reviewed by a native English speaker.

Author Response

Responses to Reviewer 1:

We appreciate your attention to our manuscript. We feel that our submission has been considerably enhanced by implementing your valuable advice. We addressed the specific comments as outlined in detail below, and modifications to the revised text are highlighted in red. We hope you will agree that the changes have rendered our report acceptable for publication.

This study conducted on a small group of subjects is interesting and quite original. However, this reviewer raises some issues that have to be addressed by authors.

Major comments

1- Authors must add diabetes in the exclusion criteria. They probably did not do so by considering it implicit.

Response

We added diabetes to our exclusion criteria.

2- Moreover, if the patients were not diabetic, were all ADA/EASD diagnostic criteria used to rule out diabetes?

Response

We used all of the ADA/EASD diagnostic criteria to rule out diabetes.

3- The sample size of this study is very small indeed. This issue, in addition to being stressed perhaps with more emphasis in the limitations section of the study, should also be adequately commented at the beginning of the discussion.

Response

We have highlighted the small number of subjects in the section on limitations of the study, and in addition have commented appropriately at the beginning of the discussion.

4- Correctly, the authors stated ‘Systemic arterial stiffness is known to increase during acute hyperglycemia’ as well as ‘Arterial stiffness is an important determinant of cardiovascular risk’. Actually, hyperglycemia "per se" is a condition that worsens CV outcome during ACS in individuals with no known diabetes (Journal of Clinical Endocrinology and Metabolism Volume 97, Issue 3, March 2012, Pages 933-942. doi: 10.1210/jc.2011-2037 - Journal of Diabetes Research, 2018, art. no. 3106056. doi: 10.1155/2018/3106056). Both this clinically very important issue and above referenced papers should be commented upon in the discussion.

Response

We have revised the discussion as suggested.

Minor comments

1- Figure 1 is too small and hard to read.

Response

We have made Figure 1 larger and easier to read.

2- The manuscript should be reviewed by a native English speaker.

Response

We have our manuscripts reviewed by native English speakers.

Reviewer 2 Report

  1. Check the title again. The words “frequency” instead of the number, and “female” must be added to your title.

Abstract.

Add a sentence to show the background of literature for your topic. Also, in results add some of the statistical indexes (ex. p < 0.05, etc.).

  1. The aim of the study was to examine or evaluate

Introduction

31 - 33. Please, rephrase it. It is not clear what you want to say.

47 – 48. This could be your general sentence and to put it at the beginning of the paragraph. Then, describe, the effective duration values of exercise as you found in the literature.

  1. “lower” or “higher”? Please, check your sentence.
  2. Please, write it clearer.

It will be better to start with something more general such as exercise and its effects on health. Then, to talk about the contribution of exercise on arterial stiffness in hyperglycemic state and to close the introduction with the frequency of exercise which is your topic. With this structure in the introduction, the purpose of your study will be clearer.

Methods

Generally, the flow of your information must start with the number of participants who were calculated for the study, then the legibility criteria, and the number of participants with demographics.

85-88. It is better to start with inclusion criteria and to write after the exclusion criteria which was the non – inclusion or something else. Also, did you compute the number of the participants who were needed to find out significant difference?

  1. Your participants were only female. In your study you talk generally about middle-aged and elderly patients. Does gender play an important role in your results? Probably, even the title must be more specific adding “female”. Also, maybe is a study limitation.
  2. Add some demographic characteristics (age, height, weight, etc.)

90-93. This information must be included in “study design”.

95-96. This information must be included at the beginning of the paragraph in “inclusion” and “exclusion criteria”.

  1. The comparison with men must be according to the literature, right? If yes, add it.
  2. You explain SBP in the row 160. Firstly, explain it in 99. Then use only the abbreviation.
  3. Start your “methodology” by talking about the way you determine the number of participants. Thus, add it in the “participants” section.
  4. Which were the criteria for the split of the participants in groups?
  5. Try to make “Figure 1” larger.
  6. Change the direction of the arrow. The intervention process goes to the right of your figure, not to the left.
  7. Can you explain how did you calculate the HRmax?
  8. Did you use any normality test (Shapiro – Wilk)? Also, did you check for Homogeneity and Sphericity violation? Then, your parametric analysis (if there was normality in your independent variables) is Two Way ANOVA with repeated measures (time*group) for your three measurements which were conducted.
  9. Write: Statistical significance was set at α = 0.05. Without the other sentence.
  10. Ethics must be a part of “Participants section”.

Results

  1. I suggest to move the “Table 1” under 3.1 section and in “Methodology” write the demographics in the text, as I wrote for line 89.

In “Table 3” the results are difficult to be read.

294-296. Write that in both groups in baseline there is a statistically significant difference at 30 and 60’. Avoid to write the same thing for the two groups.

313-316. The same suggestion as above.

Discussion

  1. Can you expose a result like this? Your sample was only female middle-aged and elderly patients.

Author Response

Responses to Reviewer 2:

We appreciate your attention to our manuscript. We feel that our submission has been considerably enhanced by implementing your valuable advice. We addressed the specific comments as outlined in detail below, and modifications to the revised text are highlighted in red. We hope you will agree that the changes have rendered our report acceptable for publication.

  1. Check the title again. The words “frequency” instead of the number, and “female” must be added to your title.

Response

We added the words "frequency" and "women" to our title.

Abstract.

Add a sentence to show the background of literature for your topic. Also, in results add some of the statistical indexes (ex. p < 0.05, etc.).

Response

We have added as you suggested.

  1. The aim of the study was to examine or evaluate

Response

We have revised it as you suggested.

Introduction

31 - 33. Please, rephrase it. It is not clear what you want to say.

Response

We have revised it as you suggested.

47 – 48. This could be your general sentence and to put it at the beginning of the paragraph. Then, describe, the effective duration values of exercise as you found in the literature.

Response

We have revised it as you suggested.

  1. “lower” or “higher”? Please, check your sentence.

Response

We have checked and corrected the text.

  1. Please, write it clearer.

It will be better to start with something more general such as exercise and its effects on health. Then, to talk about the contribution of exercise on arterial stiffness in hyperglycemic state and to close the introduction with the frequency of exercise which is your topic. With this structure in the introduction, the purpose of your study will be clearer.

Response

We have revised it as you suggested.

Methods

Generally, the flow of your information must start with the number of participants who were calculated for the study, then the legibility criteria, and the number of participants with demographics.

Response

We have revised it as you suggested.

85-88. It is better to start with inclusion criteria and to write after the exclusion criteria which was the non – inclusion or something else. Also, did you compute the number of the participants who were needed to find out significant difference?

Response

We have revised it as you suggested.

We calculated the number of subjects needed to make a significant difference.

  1. Your participants were only female. In your study you talk generally about middle-aged and elderly patients. Does gender play an important role in your results? Probably, even the title must be more specific adding “female”. Also, maybe is a study limitation.

Response

We added to the limits of our research that we targeted older women.

  1. Add some demographic characteristics (age, height, weight, etc.) 90-93. This information must be included in “study design”.

Response

We added demographic characteristics (age, height, weight, etc.) to the study design.

95-96. This information must be included at the beginning of the paragraph in “inclusion” and “exclusion criteria”.

Response

We have revised it as suggested.

  1. The comparison with men must be according to the literature, right? If yes, add it.

Response

We have revised it as suggested.

  1. You explain SBP in the row 160. Firstly, explain it in 99. Then use only the abbreviation.

Response

We have revised it as suggested.

  1. Start your “methodology” by talking about the way you determine the number of participants. Thus, add it in the “participants” section.

Response

We have revised it as suggested.

  1. Which were the criteria for the split of the participants in groups?

Response

They were randomly divided into two groups by a random number generated by a computer with a one-in-two probability.

  1. Try to make “Figure 1” larger.

Response

We have enlarged Figure 1.

  1. Change the direction of the arrow. The intervention process goes to the right of your figure, not to the left.

Response

We have revised it as suggested.

  1. Can you explain how did you calculate the HRmax?

Response

The estimated maximum heart rate based on age was calculated by subtracting age from the constant 220 and was set to 65% exercise intensity by the Carbonen method.

  1. Did you use any normality test (Shapiro – Wilk)? Also, did you check for Homogeneity and Sphericity violation? Then, your parametric analysis (if there was normality in your independent variables) is Two Way ANOVA with repeated measures (time*group) for your three measurements which were conducted.

Response

We checked for violations of the normality test (Shapiro-Wilk), homogeneity and sphericity.

  1. Write: Statistical significance was set at α = 0.05. Without the other sentence.

Response

We have revised it as suggested.

  1. Ethics must be a part of “Participants section”.

Response

We have revised it as suggested.

Results

  1. I suggest to move the “Table 1” under 3.1 section and in “Methodology” write the demographics in the text, as I wrote for line 89.

Response

We have revised it as suggested.

In “Table 3” the results are difficult to be read.

Response

We have revised it as suggested.

294-296. Write that in both groups in baseline there is a statistically significant difference at 30 and 60’. Avoid to write the same thing for the two groups.

Response

We have shown detailed results for each group.

313-316. The same suggestion as above.

Response

We have shown detailed results for each group.

Discussion

  1. Can you expose a result like this? Your sample was only female middle-aged and elderly patients.

Response

Because arterial stiffness increases with age in older postmenopausal women, there is a significant association between arterial stiffness and postprandial blood glucose levels [Gynecol. Obstet. Invest. 2005, 60, 162–166, doi:10.1159/000086570]. In fact, Tsuboi et al. [Intern. Med. Tokyo Jpn. 2015, 54, 1961–1969, doi:10.2169/internalmedicine.54.3596] reported that blood glucose measured after a meal was associated with arterial stiffness in women aged 50-85 years, but not in women younger than 50 years. Therefore, as Japan is a super-aging society [Geriatr. Gerontol. Int. 2015, 15, 673–687, doi:10.1111/ggi.12450] and since arterial stiffness is higher in women than in men [Hypertens. Dallas Tex 1979 2012, 60, 362–368, doi:10.1161/HYPERTENSIONAHA.112.191148], it is important to suppress the progression of arteriosclerosis during acute hyperglycemia in the elderly female.

Round 2

Reviewer 1 Report

The authors addressed almost all issues raised by this reviewer. 

There remains only one issue that needs to be addressed.

Lines 93-95 contain an error and are poorly written. Clarify that diabetes is an exclusion criterion, or, in other words, that among the inclusion criteria it was necessary to be non-diabetic (nor prediabetic) according to the international ADA / EASD guidelines. As it is written, diabetes is incorrectly included among the inclusion criteria. Replace 'participation criteria' with 'inclusion criteria'. Please, rewrite the paragraph.

Author Response

Responses to Reviewer 1:

We appreciate your attention to our manuscript. We feel that our submission has been considerably enhanced by implementing your valuable advice. We addressed the specific comments as outlined in detail below, and modifications to the revised text are highlighted in red. We hope you will agree that the changes have rendered our report acceptable for publication.

Lines 93-95 contain an error and are poorly written. Clarify that diabetes is an exclusion criterion, or, in other words, that among the inclusion criteria it was necessary to be non-diabetic (nor prediabetic) according to the international ADA / EASD guidelines. As it is written, diabetes is incorrectly included among the inclusion criteria. Replace 'participation criteria' with 'inclusion criteria'. Please, rewrite the paragraph.

Response

We have added as you suggested.

Reviewer 2 Report

Nice work with the corrections. Please, you must check again the English in the whole manuscript.  

Author Response

Responses to Reviewer 2:

We appreciate your attention to our manuscript. We feel that our submission has been considerably enhanced by implementing your valuable advice. We addressed the specific comments as outlined in detail below, and modifications to the revised text are highlighted in red. We hope you will agree that the changes have rendered our report acceptable for publication.

Nice work with the corrections. Please, you must check again the English in the whole manuscript.

Response

We did proofreading of the English text.